# PAIREVAL: Open-domain Dialogue Evaluation with Pairwise Comparison

**ChaeHun Park    Minseok Choi    Dohyun Lee    Jaegul Choo**
KAIST AI
{ddehun,minseok.choi,aiclaudev,jchoo}@kaist.ac.kr

## Abstract

Building a reliable and automated evaluation metric is a necessary but challenging problem for open-domain dialogue systems. Recent studies proposed evaluation metrics that assess generated responses by considering their relevance to previous dialogue histories. Although effective, these metrics evaluate individual responses directly rather than considering their relative quality compared to other responses. To handle this, we propose PAIREVAL, a novel dialogue evaluation metric for assessing responses by comparing their quality against responses in different conversations. PAIREVAL is built on top of open-sourced and moderate-size language models, and we make them specialized in pairwise comparison between dialogue responses. Extensive experiments on multiple benchmarks demonstrate that our metric exhibits a higher correlation with human judgments than baseline metrics. We also find that the proposed comparative metric is more robust in detecting common failures from open-domain dialogue systems, including repetition and speaker insensitivity.[1]

## 1 Introduction

Open-domain dialogue systems aim to interact with users by generating natural and engaging responses for a given dialogue history. In this field, building an accurate automatic evaluation metric to judge the quality of such generative systems is an important but challenging task (Liu et al., 2016). The difficulties partly originate from the *one-to-many* nature of daily conversations, where a single conversation can be continued with many different follow-up utterances. This variety in dialogues makes traditional *reference-based* metrics, which compare generated responses to a limited set of known answers (i.e., references), do not correlate well with human judgments (Liu et al., 2016). The lack of reliable evaluation metrics impedes real-world deployment of open-domain dialogue systems.

Researchers have proposed various evaluation metrics to achieve reliable assessments of dialogue systems. Initial studies enhance reference-based metrics by adopting distributed and contextualized representations by neural networks (Liu et al., 2016; Zhang et al., 2019). Subsequent studies introduce *reference-free* metrics that employ prediction models to assess responses based on their relevance to the previous dialogue context (Tao et al., 2018; Lowe et al., 2017; Mehri & Eskenazi, 2020a; Zhang et al., 2021; Zhong et al., 2022), showing a higher correlation with human judgments. Moreover, prompting-based metrics have been presented to properly transfer the instruction-following and zero-shot capabilities of large language models (LLMs) for dialogue evaluation (Zhang et al., 2023; Lin & Chen, 2023).

We hold that the dialogue evaluation should aim to assign differentiated scores to responses by considering their relative quality. In other words, evaluation metrics should make calibrated scores such that they are appropriately aligned with human evaluations. Numerous correlation metrics used in the meta-evaluation of evaluation metrics like Spearman (Zar, 2005) or Kendall rank correlations (Kendall, 1955) reflect this intuition. Therefore, we argue that assessing the target responses by considering their relative appropriateness with other

---

[1]The code and models are available at https://github.com/ddehun/PairEval.

ones is a meaningful process to make more reliable evaluation metrics. In this regard, several studies evaluate responses by considering their relative quality (Sato et al., 2020; Liusie et al., 2024). However, these approaches usually require exhaustive comparison operations or human-crafted candidate responses, which may not always be available.

In this paper, we propose PAIREVAL, a novel open-domain dialogue evaluation metric with comparative assessments. Our metric assesses the individual responses by comparing their quality against only a limited number of comparison responses. Instead of relying on a commercial or proprietary LLM, our metric is built on top of a moderate-size and open-sourced LLM (Touvron et al., 2023). To elicit the comparative ability of LMs, we devise a simple but effective learning strategy with a public dialogue corpus. Experiments on multiple benchmarks show that PAIREVAL outperforms previous evaluation metrics, and sometimes even shows higher performance than metrics with a powerful proprietary LLM. Further analysis demonstrates that the pairwise evaluation approach is robust and effective in capturing common failures (e.g., repetitive outcomes) in dialogue systems.

## 2   Related Work

**Evaluation Metrics for Open-domain Dialogue Systems**   Traditional metrics like BLEU (Papineni et al., 2002) or ROUGE (Lin, 2004) that measure the N-gram overlap between generated responses and answers show a low correlation with human judgments (Liu et al., 2016). Several studies use embedding models to consider the semantic similarity between candidate responses and answers (Liu et al., 2016; Zhang et al., 2019). However, these reference-based metrics rely on a set of known answers for similarity comparison, making it hard to consider the wide semantic space of follow-up utterances for a single conversation. To tackle this problem, recent studies proposed numerous reference-free metrics that directly predict the relevance of a generated response to the given dialogue history (Tao et al., 2018; Mehri & Eskenazi, 2020a;b; Zhong et al., 2022; De Bruyn et al., 2022). Specifically, neural classification or regression models are usually trained to distinguish relevant responses from irrelevant ones. These predictive metrics have shown meaningful progress along with pre-trained language models (Ghazarian et al., 2019; Mehri & Eskenazi, 2020a; Zhong et al., 2022), data augmentation strategies (Gupta et al., 2021; Park et al., 2021), and advanced training algorithms (Huang et al., 2020; Zhang et al., 2021). In contrast, our work focuses on a comparative evaluation to consider the relative quality between responses for a more reliable evaluation. In open-ended text generation tasks, Pillutla et al. (2021) propose a corpus-level evaluation metric, namely MAUVE, which compares model-generated text distributions with human-written ones using divergence frontiers. To the best of our knowledge, Liusie et al. (2024) propose a pioneering step that introduces a comparison-based evaluation approach for various natural language generation tasks, including an open-domain dialogue generation task. However, they find that exhaustive comparison between candidate responses is needed for reliable quality estimation of dialogues, resulting in an undesirable computational overhead. Our work overcomes this challenge by specializing in moderate-size LLMs for pairwise comparison between responses. Sato et al. (2020) consider the relative rank of generated responses against other false candidates, and a response selection model is employed for this purpose. Though promising, their evaluation approach requires human evaluation to filter out correct responses from a candidate pool. In contrast, we show that even a limited number of comparisons are enough to ensure a reasonable correlation with human judgment.

**Pairwise Comparison with Large Language Models**   The pairwise comparison approach has been widely explored from various perspectives, including preference learning (Fürnkranz & Hüllermeier, 2003), recommendation (Beutel et al., 2019), reinforcement learning (Xu et al., 2020), and retrieval systems (Qin et al., 2023). Recent studies that consider the human-aligned behaviors of LMs proposed to reflect human preference in the form of comparisons over multiple model generations (Ouyang et al., 2022; Bai et al., 2022). An LLM-based pairwise evaluation has also been increasingly adopted to build system-level ranking information (Boubdir et al., 2023; Zheng et al., 2024). This work evaluates individual responses by comparing their relative quality against the limited number of responses.

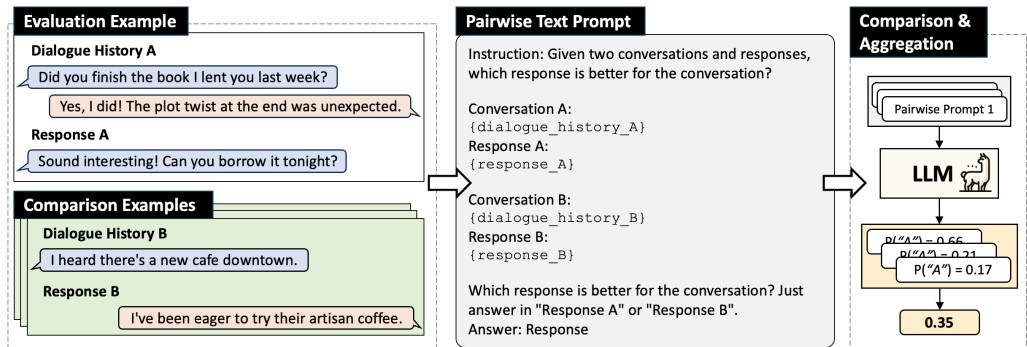

Figure 1: The overall illustration of PAIREVAL.

## 3 Method

We propose PAIREVAL, a new reference-free dialogue evaluation metric based on a pairwise comparison. Our metric assesses responses by comparing their quality against a small number of comparison examples. The comparison examples are derived from human-written conversations in a dialogue corpus. Fig. 1 depicts the overall pipeline of PAIREVAL.

### 3.1 Task Formulation

In this work, we focus on the turn-level and reference-free evaluation of open-domain dialogue systems. Given a dialogue history $h$ that consists of multiple utterances between two speakers, a dialogue system outputs a single utterance $r$ as a follow-up response. The evaluation metric $M$ considers how suitable the generated response is as the next utterance for the given dialogue history and makes an evaluation score $s = M(h, r)$. The performance of the metric is usually measured by computing the correlation between human judgments $A = \{a_1, a_2, ..., a_L\}$ and metric scores $S = \{s_1, s_2, ..., s_L\}$ when dialogue systems make $L$ individual responses $R = \{r_1, r_2, ..., r_L\}$ for their dialogue histories $H = \{h_1, h_2, ..., h_L\}$.

### 3.2 Dialogue Evaluation with Pairwise Comparison

Fig. 1 illustrates the evaluation process of PAIREVAL. The metric is designed to judge the quality of a generated response by considering its relative quality against other comparison responses. We use an LLM along with a carefully designed text prompt to perform a pairwise comparison between two responses. Given a pair of conversations in the form of a text prompt, the LM is asked to choose the better response. The probabilities of the target response being better than each comparison example are then aggregated and utilized as its final score. The generated response receives higher metric scores as it is predicted as better than the other responses. To construct a group of comparison examples, we use a few randomly sampled conversations from a public dialogue corpus. This makes our metric to be a practical and efficient solution for dialogue evaluation.

Formally, let $(h_i, r_i)$ denote a target conversation that consists of a generated response $r_i$ under evaluation and its corresponding dialogue history $h_i$. Let $C = \{(h_1^c, r_1^c), (h_2^c, r_2^c), ..., (h_N^c, r_N^c)\}$ represent $N$ different comparison examples. We first make a pair of a target conversation with every comparison example $\{(h_j^c, r_j^c)\}_{j=1}^N$, resulting in $N$ different pairs of conversations. Each pair is then converted into an input text of an LLM by replacing the placeholder of our text prompt $x_{ij} = T(h_i, r_i, h_j^c, r_j^c)$. Given the text prompt $x_{ij}$, the LM ($\theta$) is asked to choose a response of higher quality. To precisely acquire the LM's prediction in the form of predictive probability, we assign a single label word for each response (i.e., "A" and "B"), and regard the probability of each label word as a score of the response allocated by the LM. The probabilities that a generated response $r_i$ is in better quality $s_{ij} = P("r_i \text{ is better than } r_j"|x_{ij}, \theta)$ are stored during every comparison against

$c_j$. The averaged probability of $r_i$ after every competition is then used as the final evaluation score of the target response. For the aggregation of probabilities with different comparison examples, we simply use an average operation ($s_i = \sum_{j=1}^{N} s_{ij}$). In practice, since LLMs are known to be sensitive to ordering in the prompt (Wang et al., 2023), we infer the LMs twice for a single conversation pair with swapped orders and use the averaged probability.

### 3.3 Making LMs Specialized in Pairwise Comparison

Although recent LLMs have shown impressive instruction-following and task generalization abilities, applying moderate-size LLMs directly for pairwise dialogue evaluation tasks described in Section 3.2 may make it hard to ensure reasonable evaluation results. For instance, the training examples that require LMs to compare two text inputs and choose the better one would occupy only a small portion of the entire training dataset. To tackle this problem, we propose an intuitive training strategy to make LMs specialize in pairwise comparisons for evaluation.

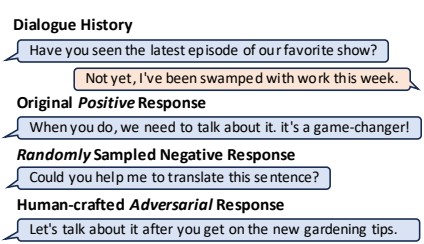

Figure 2: Responses in different types to finetune a LM in PAIREVAL.

Specifically, we construct synthetic training examples that instruct an LM to compare two responses of different quality – *positive* and *negative* responses – and choose the better one. To obtain a positive response along with its dialogue history, we simply use usual conversations that consist of a dialogue history and its follow-up utterances presented in the ordinary dialogue corpus. For reliable construction of negative ones, we explore two different types of negative responses with varied difficulties for evaluation: *random* and *adversarial*. The random negative responses are the utterances of different dialogue histories that are arbitrarily sampled from the dialogue corpus. The Adversarial negative responses are written by human annotators to exhibit high superficial relevance to a dialogue context but are not suitable as a follow-up utterance of the context. This adversarial response enables an LM to identify the inappropriateness of a response by capturing more subtle errors rather than relying on the superficial similarity against a dialogue history. In this work, we use human-written negative responses released by Sai et al. (2020) for the adversarial responses. Fig. 2 illustrates an example of responses with different types. Given a paired example with positive and negative responses, we construct an input text of an LM with pairwise evaluation prompt $T$. We then finetune the LM to predict the label word of a positive response correctly. The location of positive and negative examples in an input text is randomly decided to avoid unintended positional bias.

## 4 Experiments

The section presents our experimental setups (section 4.1), main results (section 4.2), and further analysis (section 4.3) as follows.

### 4.1 Setup

#### 4.1.1 Dataset

**Meta-Evaluation Dataset** We evaluate the performance of each evaluation metric by comparing its outcome scores against scores assigned by human annotators. To this end, multiple meta-evaluation datasets with human annotations are used for experiments. Each instance in the dataset consists of (1) dialogue history, (2) generated response from a dialogue system, (3) answer response, and (4) response quality score annotated by human annotators. The details of each meta-evaluation dataset are as follows.

*DailyDialog-Zhao* (Zhao et al., 2020) consists of 900 evaluation instances from multiple different dialogue systems. The dataset originates from the DailyDialog corpus (Li et al.,

2017), serving both as a training corpus of dialogue systems and the source of dialogue histories. The "overall" aspect of the example is used as a human annotation score. The inter-annotator agreement is measured by Kripendorff's alpha, with a result of 0.8. *ConvAI2-USR* and *TopicalChat-USR* are datasets released by Mehri & Eskenazi (2020a) and consist of 300 and 360 instances, respectively. Each dataset is derived by ConvAI2 (Dinan et al., 2020) and TopicalChat (Gopalakrishnan et al., 2019) dialogue corpus, respectively. Three dialogue systems are used as a dialogue system, and the "Overall Quality" score is used in our experiments. The inter-annotator agreements are measured by Spearman correlation with a result of 0.66 and 0.72 for each dataset. *DailyDialog-GRADE* and *ConvAI2-GRADE* are released by Huang et al. (2020), each based on the DailyDialog and ConvAI2 datasets. DailyDialog-GRADE and ConvAI2-GRADE both contain 300 instances from two and four systems, respectively. FED (Mehri & Eskenazi, 2020b) contains 375 instances of conversations between human speakers or human and dialogue systems. The "Overall" aspect of the examples is used for evaluation. The inter-annotator agreement is measured by Spearman correlation with a result of 0.82.

**Training Dataset**   We finetune an LLM with two widely used open-domain dialogue corpus - *DailyDialog* (Li et al., 2017) and *ConvAI2* (Dinan et al., 2020) - individually, resulting in two different versions of PAIREVAL. Both corpora contain multi-turn conversations between two speakers, and the number of conversations is 13,118 and 17,818, respectively. To construct positive response sets, we use randomly sampled utterances along with their previous dialogue history in the train split of a dialogue corpus. For random negative response sets, we use randomly sampled utterances from a dialogue corpus. During the training of the LM on the DailyDialog corpus, we also use human-written adversarial responses released by Sai et al. (2020) as an *adversarial* negative type. The number of training examples is set to 80k and 65k for DailyDialog and ConvAI, respectively. The LM trained on the ConvAI2 dataset is used for the evaluation of *ConvAI2-GRADE* and *ConvAI2-USR*. The LM trained on the DailyDialog dataset is evaluated on the remaining meta-evaluation datasets.

### 4.1.2  Baselines

The following baseline metrics are considered in our experiments. For better readability, we group them into reference-based and reference-free metrics. Further details are in A.1.

**Reference-based Metrics**   *BLEU-2* (Papineni et al., 2002) measures the n-gram overlap similarity of a generated response against an answer (reference) response. *BERTScore* (Zhang et al., 2019) uses contextualized embeddings of a pre-trained LM (Liu et al., 2019) for similarity comparison. *BLEURT* (Sellam et al., 2020) is pre-trained on synthetic datasets for a better evaluation of machine translation systems.

**Reference-free Metrics**   *USR* (Mehri & Eskenazi, 2020a) is trained to distinguish the original follow-up response of a dialogue history from randomly sampled ones. *DEnsity* (Park et al., 2023) calculates the distance of a generated response from the distribution of relevant responses on the feature space of an LM. *FED* (Mehri & Eskenazi, 2020b) and *FULL* (De Bruyn et al., 2022) both measure the likelihood of predefined follow-up utterances to estimate the quality of a generated response. *UniEval* (Zhong et al., 2022) adopt intermediate training on multiple datasets for an evaluation of various natural language generation tasks. *LLMEval* (Lin & Chen, 2023) is a prompt-based metric that leverages a proprietary LLM (`claude-v1.3`) for a dialogue evaluation task.

Besides, we also introduce DIRECTEVAL, which uses the same training examples and the LLM as PAIREVAL. Instead of performing a pairwise comparison, however, this metric directly predicts the appropriateness of a generated response. Specifically, we ask the LM to predict the quality of a generated response (*"Is the above response a good response to the given conversation?"*), following previous studies (Gupta et al., 2022; Liang et al., 2023). The probability of a target response ($P("Yes")$) is then used as a metric score. This metric is designed to validate the effectiveness of pairwise comparison by controlling the training configuration to be roughly the same as PairEval.

| Methods | DailyDialog GRADE | | DailyDialog Zhao | | ConvAI2 GRADE | | ConvAI2 USR | | TopicalChat USR | | FED | | Avg. | |
|---|---|---|---|---|---|---|---|---|---|---|---|---|---|---|
| | r | ρ | r | ρ | r | ρ | r | ρ | r | ρ | r | ρ | r | ρ |
| *Reference-based Metrics* | | | | | | | | | | | | | | |
| BLEU-2 (Papineni et al., 2002) | 14.6* | 10.3* | 35.5 | 20.6 | 5.41* | 9.64* | 32.3 | 31.5 | 45.9 | 46.2 | - | - | - | - |
| BLEURT (Sellam et al., 2020) | 17.5* | 12.2* | 34.1 | 28.7 | 16.9* | 16.9* | 33.3 | 29.9 | 44.7 | 41.7 | - | - | - | - |
| BERTScore Zhang et al. (2019) | 12.9* | 9.9* | 36.4 | 29.4 | 26.0 | 27.7 | 33.3 | 28.3 | 46.9 | 46.2 | - | - | - | - |
| *Reference-free Metrics* | | | | | | | | | | | | | | |
| FED (Mehri & Eskenazi, 2020b) | 2.6* | 0.1* | -9.0* | -8.5* | -6.1* | -4.6* | -2.0* | -0.7* | -7.1* | -6.9* | 11.9* | 9.4* | -1.6 | -1.9 |
| USR (Mehri & Eskenazi, 2020a) | 27.5 | 23.8 | 48.8 | 51.6 | 40.3 | 40.0 | 60.9 | 48.1 | 40.7 | 32.5 | 11.4 | 11.7 | 38.2 | 34.6 |
| CTC (Deng et al., 2021) | -15.1* | -14.7* | -5.6* | -9.2* | 3.9* | 4.1* | 47.4 | 48.3 | 39.8 | 36.3 | 16.1* | 19.5* | 14.4 | 14.0 |
| FULL (De Bruyn et al., 2022) | -8.8* | -10.0* | 5.1* | 6.0* | 22.0* | 23.8* | 7.0* | 8.9* | -4.9* | -7.2* | 47.1 | 50.6 | 11.3 | 12.0 |
| UniEval (Zhong et al., 2022) | 7.6* | 5.7* | 30.9 | 30.1 | 46.6 | 47.9 | 65.1 | 65.9 | 62.0 | 64.5 | 29.8 | 29.9 | 46.9 | 47.7 |
| DEnsity (Park et al., 2023) | 30.3 | 29.5 | 56.8 | 57.0 | 48.0 | 48.6 | 57.0 | 63.0 | 16.3 | 24.7 | 24.5 | 21.4 | 38.8 | 40.7 |
| DIRECTEVAL | 38.6 | 45.1 | 62.4 | 70.0 | 55.5 | 56.2 | 58.3 | **75.0** | 39.8 | **73.9** | 44.4 | 50.4 | 49.8 | 61.8 |
| PAIREVAL (Ours) | **47.8** | **55.8** | **62.9** | **70.7** | 51.1 | 56.6 | **66.5** | 71.1 | **70.8** | 72.2 | 51.9 | **52.0** | **58.5** | **63.1** |
| *Proprietary LLM-based Metrics* | | | | | | | | | | | | | | |
| LLMEval (Lin & Chen, 2023) | 34.6 | 34.9 | - | - | **61.3** | **61.8** | 53.3 | 51.5 | 49.0 | 49.9 | **59.7** | 49.9 | - | - |

Table 1: The correlations between automatic metrics and humans on meta-evaluation datasets. $r$ and $\rho$ denote Pearson correlation and Spearman's rank correlation coefficient, respectively. All values with p > 1e-5 are marked with *. The highest and the second highest scores in each column are marked in **bold** and underline, respectively. For PAIREVAL, we report the averaged scores over five runs with different comparison examples.

| LM | DailyDialog Zhao | | DailyDialog GRADE | | TopicalChat USR | | FED | |
|---|---|---|---|---|---|---|---|---|
| | $\mu$ | $\sigma$ | $\mu$ | $\sigma$ | $\mu$ | $\sigma$ | $\mu$ | $\sigma$ |
| Zero-shot | 16.2 | 5.0 | 31.1 | 3.5 | 42.6 | 2.8 | 44.6 | 3.6 |
| Fine-tuned | 55.5 | 1.1 | 70.5 | 0.4 | 71.9 | 0.5 | 51.2 | 1.1 |

Table 2: Mean ($\mu$) and standard deviation ($\sigma$) results of fifteen runs with different random comparison examples ($N = 1$).

### 4.1.3 Implementation Details

We use `Llama-2-7b-chat` (Touvron et al., 2023) as an LM of both PAIREVAL and DIRECTEVAL. The LM is finetuned with LoRA (Hu et al., 2021) for 1 epoch, and the $r$ and $\alpha$ of LoRA is set to 16 and 8, respectively. AdamW (Loshchilov & Hutter, 2018) optimizer is used for optimization, and the initial learning rate is set to 1e-4. The batch size is set to 16. The number of comparison examples ($N$) is set to 3 and is randomly sampled from the validation set of the DailyDialog corpus. Note that these comparison examples are used across all meta-evaluation datasets. In all experiments, a single 3090 RTX GPU with 24GB of memory is used. Regarding the limited computational resources, we use 4-bit quantization (Dettmers et al., 2024) during the evaluation of DIRECTEVAL and PAIREVAL.

## 4.2 Main Results

We use the Pearson correlation coefficient ($r$) and Spearman's rank correlation coefficient ($\rho$) to measure the correlation between human judgments and metric scores. Table 1 shows the experimental results of all metrics. Overall, PAIREVAL achieves the highest performance in four and three out of six datasets in the Pearson and Spearman correlations, respectively. Our metric works well on dialogues from TopicalChat-USR and FED that are not used during the finetuning stages. Moreover, PAIREVAL sometimes even shows a higher correlation than LLMEval which relies on a powerful proprietary LLM. Among the other baseline metrics, UniEval shows reasonable performance. DIRECTEVAL, a metric that uses the same training dataset source as PAIREVAL, also shows competitive performance across multiple benchmarks. However, it generally performs worse than PAIREVAL, especially in the case of the Pearson correlation coefficient. We believe these results confirm the effectiveness of the pairwise comparison approach for dialogue evaluation.

## 4.3 Analysis

We conduct further analysis to understand PAIREVAL comprehensively as follows.

| $N$ | DailyDialog GRADE | | DailyDialog Zhao | | ConvAI2 GRADE | | ConvAI2 USR | | TopicalChat USR | | FED | |
|---|---|---|---|---|---|---|---|---|---|---|---|---|
| | $r$ | $\rho$ | $r$ | $\rho$ | $r$ | $\rho$ | $r$ | $\rho$ | $r$ | $\rho$ | $r$ | $\rho$ |
| *Comparison Examples from Dialogue Corpus (Random)* | | | | | | | | | | | | |
| $N=1$ | 42.2 | 56.1 | 57.0 | 70.6 | 48.3 | 56.5 | 61.3 | 70.9 | 64.5 | 72.1 | 46.2 | 50.9 |
| $N=2$ | 45.5 | 56.1 | 60.8 | 70.7 | 49.0 | 56.3 | 66.3 | 71.1 | 69.9 | 72.3 | 50.6 | **52.0** |
| $N=3$ | 47.8 | 55.8 | 62.9 | 70.7 | 51.1 | **56.6** | **66.5** | 71.1 | 70.8 | 72.2 | 51.9 | **52.0** |
| *Comparison Examples from Meta-Evaluation Dataset (Test)* | | | | | | | | | | | | |
| $N=1$ | 50.9 | 55.8 | 57.3 | 70.3 | 46.7 | 55.8 | 51.9 | 71.5 | 60.1 | 72.1 | 44.5 | 49.9 |
| $N=2$ | 54.1 | 56.4 | 65.3 | **71.0** | 50.2 | **56.6** | 63.7 | **72.0** | 70.1 | 72.7 | 51.1 | 51.0 |
| $N=3$ | **54.9** | **56.5** | **67.1** | **71.0** | **53.9** | 56.2 | 65.9 | 71.7 | 69.9 | **72.8** | 50.9 | 51.4 |

Table 3: Results with changed number ($N$) and type (*Random* and *Test*) of comparison examples. The highest score within the same type of comparison example is underlined. The highest score among all configurations is further highlighted in **bold**. All results are averaged scores over five runs with different comparison examples.

| Methods | DailyDialog GRADE | | ConvAI2 USR | |
|---|---|---|---|---|
| | $r$ | $\rho$ | $r$ | $\rho$ |
| Zero-shot | 16.6 | 17.3 | 30.1 | 29.8 |
| Fine-tuned | 55.8 | 56.4 | 70.0 | 71.9 |

Table 4: Results with an exhaustive comparison between meta-evaluation examples.

### 4.3.1 Impacts of Finetuning on Correlation and Stability of PairEval

To probe the validity of finetuning described in Section 3.3, we replace the finetuned LM in PAIREVAL with a pretrained one and observe the results. For experiments, we use a randomly sampled single conversation from a DailyDialog validation split as a comparison example. We report the mean and standard deviation of multiple runs (15) with different comparison examples. As shown in Table 2, finetuning on pairwise evaluation examples greatly contributes to the performance of PAIREVAL. Furthermore, PAIREVAL with the finetuned LM is more robust to the changes of comparison examples, confirming its validity.

### 4.3.2 Changed Number and Types of Comparison Examples

We next analyze the impacts of different configurations for comparison examples in PAIREVAL by (1) reducing the number of comparison examples ($N$) and (2) using a few examples in the target meta-evaluation dataset (*Test*) instead of a dialogue corpus (*Random*). We report the averaged results of five runs with different comparison examples. Finetuned LMs are used for the experiments, and results are shown in Table 3. We first observe that the increased number of comparison examples usually contributes to a better correlation with human judgments. Regarding the different types of comparison examples, using the examples in the meta-evaluation datasets usually offers better performance. However, access to such test examples may not always be available. Therefore, we believe that using randomly sampled conversations from a dialogue corpus can be a reasonable option.

We also explore the more exhaustive evaluation case, where a single evaluation sample is compared against all other samples in a meta-evaluation dataset to induce the final score. In other words, if we have $M$ different responses under evaluation, the number of an LM inference becomes $\mathcal{O}(M^2)$. Results in Table 4 show that conducting an exhaustive comparison between all evaluation examples contributes to the increased correlation. For instance, In the ConvAI2-USR dataset, the correlations of PAIREVAL are increased from 66.5 and 71.1 to 70.0 and 71.9 in Pearson and Spearman correlation, respectively. Nevertheless, when considering the efficiency of an evaluation process, relying on a few ($N \leq 3$) comparison examples can be a reasonable choice.

| LM | Position | DailyDialog Zhao | | DailyDialog GRADE | | TopicalChat USR | | FED | |
|---|---|---|---|---|---|---|---|---|---|
| | | $r$ | $\rho$ | $r$ | $\rho$ | $r$ | $\rho$ | $r$ | $\rho$ |
| Zero-shot | First | 7.4 | 7.2 | 24.2 | 27.1 | 33.0 | 35.5 | 41.4 | 47.0 |
| | Second | 15.4 | 19.4 | 26.5 | 30.2 | 35.4 | 40.0 | 23.3 | 33.5 |
| | Both | 14.3 | 15.9 | 31.2 | 32.2 | 39.8 | 42.2 | 42.3 | 44.5 |
| Finetuned | First | 40.8 | 54.3 | 55.8 | 69.5 | 57.7 | 73.2 | 45.3 | 50.6 |
| | Second | 40.9 | 56.8 | 54.8 | 70.8 | 62.6 | 70.6 | 45.4 | 50.1 |
| | Both | 42.2 | 56.1 | 57.0 | 70.6 | 64.5 | 72.1 | 46.2 | 50.9 |

Table 5: Analysis of position bias results with a single random comparison example ($N = 1$). The *Position* denotes a location of evaluation examples in an input prompt. We report the averaged results of five runs with different comparison examples.

| Comparison Example | Hard Negative | DailyDialog GRADE | | DailyDialog Zhao | | TopicalChat USR | | FED | |
|---|---|---|---|---|---|---|---|---|---|
| | | $r$ | $\rho$ | $r$ | $\rho$ | $r$ | $\rho$ | $r$ | $\rho$ |
| Random | | 36.9 | 39.6 | 62.4 | 69.2 | 69.0 | 69.9 | 45.3 | 45.0 |
| | ✓ | 50.8 | 56.4 | 66.4 | 70.7 | 68.9 | 72.4 | 52.4 | 51.6 |
| Test | | 37.5 | 38.2 | 63.8 | 68.7 | 70.9 | 72.5 | 43.4 | 43.4 |
| | ✓ | 49.0 | 55.5 | 64.5 | 71.1 | 71.1 | 71.8 | 53.3 | 51.8 |

Table 6: Results with ablating human-written hard negative for LM finetuning ($N = 1$). The *Test* denotes comparison examples from each meta-evaluation dataset.

### 4.3.3 Position Bias

Recent studies have reported that an LLM-based comparative evaluation usually suffers from positional bias, where an LLM exhibits spurious preference over instances in a certain position in the input texts (Wang et al., 2023; Zheng et al., 2024). To analyze the impacts of such bias on performance, we locate the target evaluation conversation in the *First* and *Second* positions of an input prompt respectively, and observe the changed performance. Besides, we also report the original scores of PAIREVAL (*Both*) that use an averaged probability when the target response is located in *First* and *Second* positions. A single conversation is used as a comparison example for experiments. As shown in Table 5, both zero-shot and finetuned LMs usually suffer from position bias to some extent. This problem is more significant in a zero-shot setting, where the max value of a correlation difference between the *First* and *Second* is 18.1 (41.4 and 23.3 in FED). The finetuned LM looks relatively free from such issues, where a maximum correlation difference is 4.9 (57.7 and 62.6 in TopicalChat-USR).

### 4.3.4 Impacts of Human-Written Adversarial Negatives for LM Finetuning

We verify the effectiveness of human-written adversarial examples described in Section 3.3. Table 6 presents the results of PAIREVAL when the LM is finetuned with and without human-written hard negative responses. For a fair comparison, we use the same number of training examples for both cases by replacing the hard negatives with random ones. From the results, we confirm that adversarial examples generally contribute to increasing the correlation of PAIREVAL with human evaluation. One remaining issue is that the creation process of adversarial responses involves a human annotation process, which is not always available and is not scalable. In this regard, a dataset generation strategy with LLMs can be an efficient and effective alternative (Yoo et al., 2021; Liu et al., 2022; Lee et al., 2022). We leave such explorations as our future work.

### 4.3.5 Robustness to Adversarially Manipulated Responses

Khalid & Lee (2022) report that automated metrics for dialogue evaluation often struggle to give higher scores to correct responses rather than to adversarially curated and incorrect ones. We probe whether our pairwise comparative approach can alleviate such issues and can accurately identify subtly corrupted responses. To this end, we use a meta-evaluation

| Method | Adversarial Attack Type | | | | |
|---|---|---|---|---|---|
| | Entailment | Pronoun Usage | Named Entities | Speaker Sensitiveness | Contradict |
| DIRECTEVAL | 93.9 | 95.7 | 94.9 | 79.0 | 95.5 |
| PAIREVAL | 92.3 | 98.6 | 94.9 | 95.6 | 96.4 |

(a) Evaluation results with different adversarial attack types.

| Method | Adversarial Attack Type | | | | | |
|---|---|---|---|---|---|---|
| | Repetition | Vocabulary Diversity | Bad Paraphrase | Entrainment | Dullness | **Avg.** |
| DIRECTEVAL | 71.9 | 79.9 | 91.9 | 82.1 | 97.5 | 88.2 |
| PAIREVAL | 98.4 | 97.0 | 95.1 | 96.2 | 97.1 | 96.2 |

(b) *(cont'd)* Evaluation results with different adversarial attack types.

Table 7: **Accuracy of Different Metrics on Adversarial Attack** The accuracy is higher if the original response receives a higher score than a corrupted one by a metric. The higher score for each column is highlighted in underline.

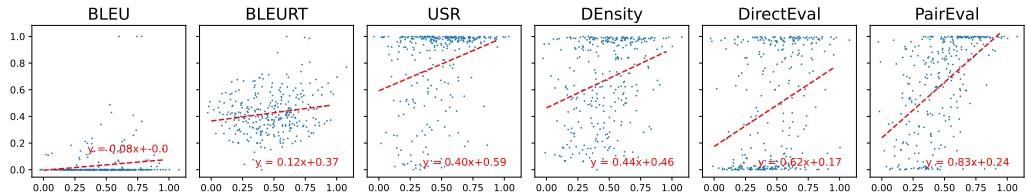

Figure 3: Scatter plots between human judgments and metric scores on the DailyDialog-Grade dataset. Each point indicates a response, and the x and y values of each point indicate denote human and metric scores, respectively. We add a noise sampled from $\mathcal{N}(0, 0.03^2)$ to human scores for better visualization. The red line indicates a linear regression.

dataset proposed by Khalid & Lee (2022). The dataset is created by manipulating the original human-written response into various corruption strategies. Each evaluation example in the dataset consists of a (1) dialogue history, (2) original response, and (3) corrupted response. The metric should assign a higher score to the original response than the corrupted one. For experiments, we compare the performance of PAIREVAL and DIRECTEVAL. PAIREVAL compares two responses – original and corrupted – by locating them in a single input text, while DIRECTEVAL evaluates two responses individually. As shown in Table 7, PAIREVAL achieves higher accuracy in most attack types, confirming its robustness on dialogue evaluation.

### 4.3.6 Qualitative Results

**Metric Visualization with Scattered Predictions** Fig. 3 presents a prediction of selected metrics along with human judgments in the DailyDialog-GRADE dataset. The x and y axes of each point represent human judgments and metric scores, respectively. Among the baseline metrics, reference-based metrics like BLEU and BLEURT fail to assign discriminated scores as human ones. Reference-free metrics like USR and DirectEval show a positive correlation with human judgments. PAIREVAL shows a strong correlation with human evaluations.

**Case Study** Table 8 presents two selected examples from meta-evaluation datasets and their evaluation results of automatic metrics and human annotators. For both examples, PAIREVAL evaluates generated responses closely to human scores.

| History | A: Children's books require less writing and more illustrations. It would be worth it for the extra 500 copies though 
 B: Haha I agree with you. Do you ever buy stuff from Amazon? do you know why Jeff Bezos put a clock into a mountain lol | | | | | | |
|---|---|---|---|---|---|---|---|
| Reference | A: I buy things from Amazon all the time. Isn't that the clock that's supposed to run for 10,000 years | | | | | | |
| Response | A: I have not, what is it about? | | | | | | |
| Evaluation | Human | BLEU-2 | BERTScore | FULL | UniEval | DirectEval | PairEval |
| Score(Rank) | 3.0(0.56) | 0.0(0.95) | 0.84(0.82) | 0.76(0.72) | 0.76(0.72) | 0.99(0.78) | 0.47(0.65) |
| History | A: What do you think about the equipment in our company? 
 B: We have to equip our company with a network of work stations at every office and show room as soon as possible. | | | | | | |
| Reference | A: I also realized this issue. And office automation is essential. | | | | | | |
| Response | A: How do you ensure quality? | | | | | | |
| Evaluation | Human | BLEU-2 | BERTScore | FULL | UniEval | DirectEval | PairEval |
| Score(Rank) | 3.9(0.09) | 0.0(1.0) | 0.86(0.36) | 0.61(0.71) | 0.61(0.71) | 0.98(0.28) | 0.33(0.2) |

Table 8: An example of TopicalChat-USR and DailyDialog-Grade datasets with evaluation results of selected metrics. We also report the rank score of each metric inside the parentheses, where the rank of the metric score is divided by the total number of evaluated examples.

## 5    Conclusion

In this work, we propose PAIREVAL, a new comparative evaluation paradigm for assessing the quality of individual responses by considering their relative quality against a few comparison examples. We encourage moderate-size and open-source LLMs to be specialized for pairwise comparison. Experiments on multiple evaluation benchmarks demonstrate that PAIREVAL correlates well with human judgments, confirming its effectiveness and validity. Although effective, PAIREVAL inevitably introduces multiple LLM calls for a single evaluation. This problem would be amplified as we use a larger set of comparison examples. Our future work should address an efficient way to find an optimal and small number of comparison examples.

## Acknowledgement

We highly appreciate the reviewers' insightful comments on our manuscript. This work was supported by Institute for Information & communications Technology Promotion(IITP) grant funded by the Korea government(MSIT) (No.RS-2019-II190075 Artificial Intelligence Graduate School Program(KAIST)) and the National Research Foundation of Korea(NRF) grant funded by the Korea government(MSIT) (No. NRF-2022R1A2B5B02001913 and No. 2022R1A5A7083908).

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

# A Appendix

## A.1 Baseline Details

We present further implementation details for baselines.

- BLEU-2 (Papineni et al., 2002): We use BLEU-2 score implemented in Natural Language Toolkit (NLTK) library (Bird et al., 2009).
- BLEURT (Sellam et al., 2020): We use an `Elron/bleurt-tiny-512` in Huggingface Hub.[2]
- BERTScore (Zhang et al., 2019): We use an official implementation[3] by the authors with RoBERTa-large (Liu et al., 2019).
- FED (Mehri & Eskenazi, 2020b): We use an official implementation[4] by the authors with DialoGPT (Zhang et al., 2020).
- CTC (Deng et al., 2021): We use official models and implementation[5] by the authors.
- USR Mehri & Eskenazi (2020a): We use USR-Retrieval for the overall experiments. For results in TopicalChat-USR and ConvAI2-USR, we cite the results in Mehri & Eskenazi (2020a). For the FED dataset, we cite results reported by Yeh et al. (2021). For other meta-evaluation datasets, we use a BERT (Devlin et al., 2019) finetuned on a response selection task.
- FULL (De Bruyn et al., 2022): We use an official implementation[6] by the authors.
- UniEval (Zhong et al., 2022): We use use official model and implementation[7] by the authors.
- DEnsity (Park et al., 2023): We use official models and implementation[8] by the authors.

## A.2 Additional Qualitative Results

We present more visualized predictions of selected metrics on other datasets in Fig. 4, Fig. 5, and Fig. 6. From the results in DailyDialog-Zhao dataset (Fig. 4), we observe that PAIREVAL often outputs predictions concentrated on a certain range (around 0.5). We believe such behaviors can be alleviated through post-hoc calibration techniques like temperature scaling (Guo et al., 2017) if necessary.

---

[2] https://huggingface.co/Elron/bleurt-tiny-512
[3] https://github.com/Tiiiger/bert_score
[4] https://github.com/Shikib/fed
[5] https://github.com/tanyuqian/ctc-gen-eval
[6] https://github.com/maximedb/full
[7] https://github.com/maszhongming/UniEval
[8] https://github.com/ddehun/DEnsity/tree/master

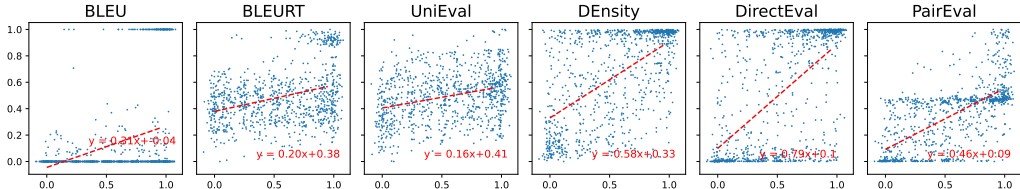

Figure 4: Scatter plots between human judgments and metric scores on the DailyDialog-Zhao dataset. The indicators are the same as Fig. 3.

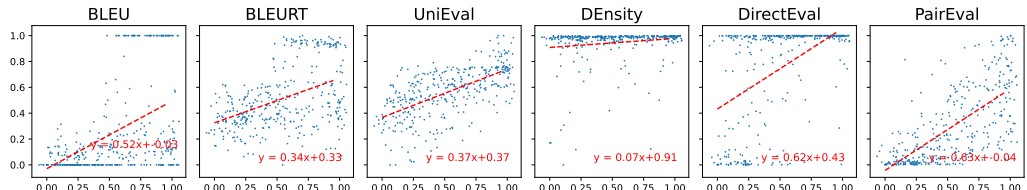

Figure 5: Scatter plots between human judgments and metric scores on the TopicalChat-USR dataset. The indicators are the same as Fig. 3.

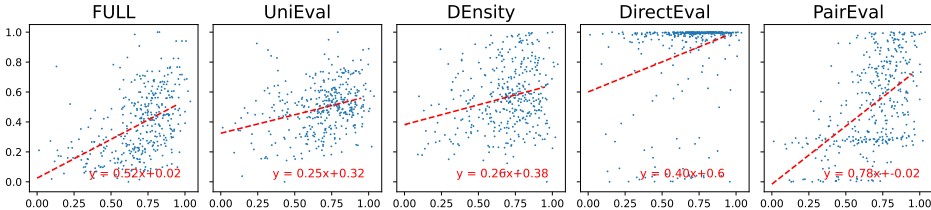

Figure 6: Scatter plots between human judgments and metric scores on the FED dataset. The indicators are the same as Fig. 3.

