# OpenReview forum: "PairEval: Open-domain Dialogue Evaluation Metric with Pairwise Comparisons"
_colmweb.org/COLM/2024/Conference — COLM_

### Official Review · Reviewer_HoEo · 2024-05-08

**Rating:** 7
**Confidence:** 3
**Ethics Flag:** 1

**Summary:**

The authors introduce PairEval a dialogue evaluation metric for assessing responses by comparing their quality against responses in different conversations. For this purpose they fine-tune a LLM to conduct the specific pair-wise dialogue evaluation task proposed. Evaluation results show better correlation with human evaluations than other metrics used in the evaluation. I think this paper presents an interesting idea and pushes forward the state of the art in open dialogue evaluations. COLM participants and the research community will benefit from the ideas and results presented here. However, I still feel additional work is needed to get a much better assessment of the benefits of the proposed technique. For instance, it would be interesting to understand the range of variability of achieved results by changing the comparative examples used in the evaluations, as well as the training data used to fine-tune the LLM. How much semantic similarity and/or topic overlap are there between the data used for fine-tuning and the evaluation data sets? Also, how human evaluations correlate to other human evaluations? I think an opportunity is missed here to understand better the complexity of the task for humans and get a better sense on how far or how close the proposed technique is bringing us to human evaluations.

**Questions To Authors:**

It would be interesting to understand the range of variability of achieved results by changing the comparative examples used in the evaluations, as well as the training data used to fine-tune the LLM. How much semantic similarity and/or topic overlap are there between the data used for fine-tuning and the evaluation data sets? Also, how human evaluations correlate to other human evaluations? I think an opportunity is missed here to understand better the complexity of the task for humans and get a better sense on how far or how close the proposed technique is bringing us to human evaluations.

**Reasons To Accept:**

I think this paper presents an interesting idea and pushes forward the state of the art in open dialogue evaluations. COLM participants and the research community will benefit from the ideas and results presented here.

**Reasons To Reject:**

I would probably want to see more exploratory analysis on the items described in the questions to authors, although I do not feel this is actually a reason to reject, given that this is a conference paper.

---

> ### Author Rebuttal · Authors · 2024-05-30
>
> We appreciate the reviewer’s thorough review and positive support. We hope our responses address all the concerns.
>
> **Question 1**
> > It would be interesting to understand the range of variability of achieved results by changing the comparative examples used in the evaluations, as well as the training data used to fine-tune the LLM.
>
> We acknowledge that it is an important analysis to investigate the changed performance of PairEval with different comparison examples. For instance, in Figure 4, PairEval can make predictions concentrated on a specific range if most evaluated responses are considered lower quality than their compared response. Our future work should present a more dedicated comparative example selection strategy to reduce the number of comparisons and increase the correlation with human judgments.
>
> **Question 2**
> > How much semantic similarity and/or topic overlap are there between the data used for fine-tuning and the evaluation data sets? Also, how human evaluations correlate to other human evaluations? I think an opportunity is missed here to understand better the complexity of the task for humans and get a better sense on how far or how close the proposed technique is bringing us to human evaluations.
>
> Among the meta-evaluation datasets used in our experiments, TopicalChat-USR and FED can be regarded as different datasets from the ones used in fine-tuning (i.e., DailyDialog and ConvAI2). Here, evaluation metrics for TopicalChat are required to consider not only a dialogue history and response but also background knowledge from external sources like Wikipedia.
> Among the meta-evaluation datasets used in our experiments, TopicalChat-USR and FED can be regarded as different from those used in fine-tuning (i.e., DailyDialog and ConvAI2). Here, evaluation metrics for TopicalChat should consider not only a dialogue history and response but also background knowledge from external sources like Wikipedia.
>
> Regarding correlations within the human annotators, we can refer to inter-annotator agreement scores reported by meta-evaluation datasets. For instance, Zhao et al. (2020) released a meta-evaluation dataset called DailyDialog-Zhao, and inter-annotator agreement is measured by Kripendorff's alpha, with a result of 0.8. We will include more details about the inter-annotator agreement scores of each dataset in our camera-ready version.

---

> > ### Comment · Reviewer_HoEo · 2024-06-06
> >
> > Thanks for your response to my comments. I am looking forward to seeing the information about inter-annotator agreement in the camera ready version of the paper, as well as future research work addressing some of the issues discussed.

---

### Official Review · Reviewer_qHB5 · 2024-05-08

**Rating:** 7
**Confidence:** 4
**Ethics Flag:** 1

**Summary:**

The authors propose a new metric to evaluate open-domain dialogue via pairwise comparisons of responses in different dialogues. They use an LLM to judge the system's response along with a number of randomly sampled utterance-response examples (with different context) and use the scores to calculate a final score for the system response.

**Questions To Authors:**

If you need to fine-tune a model for Pairwise evaluation, how different is this to reference-free / LLM as a judge approaches? I understand the need to bypass proprietary LLMs but we now have access to very powerful open-source LLMs that can be used as judges. It would be nice to see a comparison against PairEval.

How do you handle topic switches, humour, sarcasm, or other cases where the response may not seem to match but could be actually good, funny, etc?

How difficult is it to design the LLM prompt (Section 3.2)? If one needs to replace the LLM with a newer / different model, how much effort is it to re-design the prompt?

Do you think this approach could work on other domains e.g. math or coding? What adjustments would need to be made?

Given that your metric needs to be trained, have you tried using a larger LLM or training with more data?

**Reasons To Accept:**

This is an interesting methodology to judge open-ended responses that can be useful to the community

The paper is clear and easy to replicate

The metric is thoroughly evaluated including against human raters, outperforming metrics from related work (except LLMEval)

**Reasons To Reject:**

May not be very generalizable since it requires training (e.g. the authors use human-written negative responses)

---

> ### Author Rebuttal · Authors · 2024-05-30
>
> We appreciate the reviewer’s thorough review and positive support.
>
> **W1**
>
> We recognize that utilizing human-written examples can harm the adaption of our method to a new domain or dataset. To tackle this, we can consider replacing human annotators with an LLM for a more efficient data acquisition process. We also think the results on TopicalChat-USR and FED datasets in Table 1 can alleviate your concerns, as PairEval shows superior performance even when the training and evaluation datasets are not exactly matched.
>
> **Q1**
>
> We believe that our work's key contribution lies in proposing a new comparative evaluation paradigm for assessing the quality of individual responses. When determining the quality of a single target response, we propose to consider its relative quality compared to responses derived from other dialogs. Although our current work adopts a fine-tuning approach to specialize LMs for comparative operations, more powerful LMs may not require a training process.
>
> **Q2**
>
> Our manuscript mainly focuses on evaluating the overall quality of a response rather than its various aspects. Suppose we extend PairEval to assess responses in such fine-grained aspects. In that case, comparison responses of varied quality from different aspects can be utilized, followed by a proper finetuning strategy to adapt PairEval into evaluation from diverse perspectives.
>
> **Q3**
>
> We do not put much effort into crafting the LLM prompt in PairEval for a fair comparison and reproducibility. If the LLM should be changed into new ones, we believe one can use mostly the same prompt as the original one.
>
> **Q4**
>
> Although PairEval is mainly designed for open-domain dialogue evaluation, however, with proper adjustment,  we believe that our comparative evaluation approach can be extended to other tasks. To achieve this, we should define desirable characteristics in model outputs by considering the nature of the tasks of interest. For instance, in coding, one may focus on executability, accuracy, and reliability. Based on this consideration, we can construct comparative examples with varied levels of quality from diverse perspectives.
>
> **Q5**
>
> Since our manuscript focuses on the proof-of-concept of the comparative approach for dialogue evaluation, we did not conduct experiments with other LLMs or different datasets. The experiments were designed to utilize open-sourced LLMs and publicly available datasets. Our future study can be extended to leverage more powerful resources.

---

### Official Review · Reviewer_JUxr · 2024-05-10

**Rating:** 6
**Confidence:** 3
**Ethics Flag:** 1

**Summary:**

The paper proposed a method *PairEval* for dialog response evaluation.

*PairEval* lies between commonly used reference-based metrics and reference-free metrics. The core idea and novelty is to make the assessment decision (1) dependent on some baseline examples, while (2) avoiding the direct use of a reference response of the conversation of interest.
* The first design makes the assessment results more discriminative, especially when all evaluation examples use the same set of baseline examples, as is done in this work.
* The second design alleviates the influence of the one-to-many mapping problem in open-domain dialogs.

The whole idea is great. Experiment settings are mostly reasonable and comprehensive.

The only problem is that the proposed method *PairEval* does not improve much over the actual baseline *DirectEval*.
* Although the authors also compare it against a number of baseline metrics from prior works and *PairEval* is much stronger, the baseline *DirectEval* also outperforms other baselines a lot. It is fair to say that most of the improvement comes from using llama-2-7b-chat as the base model, and the gap between *PairEval* and *DirectEval* is less obvious.
* To be more specific, firstly *PairEval* and *DirectEval* have close Spearman's $\rho$ (63.1 vs. 61.8). Secondly, the larger gap of Pearson's $r$ (58.5 vs. 49.8) is likely to be largely reduced if we use another simple baseline which is a fine-tuned llama-2-7b-chat that regresses the output score.

Overall, the proposed method is promising and the results are positive, but I would expect more performance gain from the method. I suggest considering the following to make this work stronger.
* Make a more delicate design on the choice of baseline examples. Since all evaluation examples will share a small set of baseline examples, curating the baseline example set is not going to be very costly. Additionally, maybe we can get aspect-based evaluation and interpretability as a bonus by carefully choosing baseline examples.
* Include a fine-tuned regression model based on llama-2-7b-chat as a baseline.
* Analyze how the choice of evaluation dataset causes very different experimental results. For example, *PairEval* outperforms *DirectEval* significantly on *DailyDialog-GRADE* but it is close to or underperforms *DirectEval* on other datasets.

A minor suggestion is to include the results of non-exhaustive comparison in Table 4 for easy reading.

**Reasons To Accept:**

* A novel dialog response evaluation method that combines the advantages of reference-free and reference-based metrics while avoiding their demerits.
* Mostly comprehensive experiments and analyses.

**Reasons To Reject:**

* Marginal improvement over the actual good baseline (*DirectEval*).
* Lack of investigation into the performance discrepancy on different datasets.

---

> ### Author Rebuttal · Authors · 2024-05-30
>
> We appreciate the reviewer’s thorough review and positive support.
>
> **Weaknesses**
> > Marginal improvement over the actual good baseline (DirectEval). Lack of investigation into the performance discrepancy on different datasets.
>
> We recognize that the performance of PairEval and DirectEval is sometimes mixed. Furthermore, the meta-evaluation datasets used in our work have quite different characteristics, leading to varied tendencies across different metrics. We think Tables 6 and 7 in Section 4.3.5 can address your concern, where PairEval is superior to penalizing adversarial responses compared to DirectEval. Moreover, incorporating your valuable suggestions, including advanced training algorithms or comparison example selection strategies, will enhance PairEval's performance.
>
> **Suggestion 1**
> > Make a more delicate design on the choice of baseline examples. Since all evaluation examples will share a small set of baseline examples, curating the baseline example set is not going to be very costly. Additionally, maybe we can get aspect-based evaluation and interpretability as a bonus by carefully choosing baseline examples.
>
> As you pointed out, our current work does not delve into choosing the optimal comparison examples for better correlation. Instead, we conducted experiments with quite simple and easily replicable comparison example selection strategies (e.g., Random or Test in Section 4.3.2). We believe the essential future step to advance PairEval is finding better comparison examples by considering factors like target examples and LM’s initial prediction. We appreciate your insightful suggestions.
>
> **Suggestion 2**
> > Include a fine-tuned regression model based on llama-2-7b-chat as a baseline.
>
> We appreciate your dedicated comments! Similar to your suggestions, during our preliminary study, we trained LLM with different levels of appropriateness labels (i.e., positive:1, human-written adversarial: 0.5, and randomly sampled: 0), inspired by Ye et al. (2021). However, this training objective does not always show a better correlation, so we decided to use only discrete label space. Regarding these results, we suspect constructing a more nuanced label space or dynamically adapting its value with external feedback can be a meaningful direction. We acknowledge your helpful comments on improving our work.
>
> [References]
>
> Ye et al. Towards Quantifiable Dialogue Coherence Evaluation. ACL2021

---

> > ### Comment · Reviewer_JUxr · 2024-06-06
> >
> > Thank you for the response. The overall improvement is still marginal so I will keep my score unchanged.

---

### Official Review · Reviewer_P95M · 2024-05-12

**Rating:** 4
**Confidence:** 4
**Ethics Flag:** 1

**Summary:**

This paper introduces a novel dialogue evaluation metric for assessing one response's quality by comparing the target dialogue with a set of different dialogues.  Specifically, the method requires the model to compare the target dialogue with each of the dialogues in the set and use the accumulated comparison score as the final evaluation for the target dialogue.  The method is based on moderate-sized open-source language models. Experiments on multiple test sets show the effectiveness. However, some statements are confusing and there are a few questions about the method.

**Questions To Authors:**

4.1 This article uses artificially constructed adversarial data (from previous research). Does this impact the migration of the method in this article to other domains? Such as DSTC-9 data?

4.2 Apart from using a relatively large 7B model, what is the essential difference between your pairwise training method and the previous method that distinguishes positive and negative examples (such as DynaEval by Zhang et al., 2021)?

4.3 Why using an exhaustive comparison did not entail big improvements (Tables 3 and 4)?

4.4 The experimental results are different in Tables 3 and 6 when N=1. Why the differences?

**Reasons To Accept:**

2.1 The paper is generally well-written and easy to understand.

2.2 Extensive experiments are conducted and the results show the effectiveness of the method.

**Reasons To Reject:**

3.1 This paper needs to explain more deeply why this comparative method would lead to better evaluation results. Specifically, why comparing with other dialogue can help the model to evaluate dialogue quality?

3.2 On page 2, the authors state "However, they find that exhaustive comparison between candidate responses is needed for reliable quality estimation of dialogues, resulting in an undesirable computational overhead.". This is why they use moderate-sized open-source models, but the method in this paper requires fine-tuning the model and multiple calls to the model for one evaluation. The undesirable computational overhead still exists. A computation cost comparison between the LLM-based method and this paper should be conducted.

3.3 On page 2, the authors state "However, these approaches usually require exhaustive comparison operations or human-crafted candidate responses, which may not always be available.". Since you state that human-crafted data has the above drawbacks, why do you still conduct experiments with human-crafted adversarial data (from previous research)?

---

> ### Author Rebuttal · Authors · 2024-05-30
>
> We appreciate the reviewer’s thorough review and positive support.
>
> **W1**
>
> Our metric is designed on top of the assumption that relative judgments often yield more consistent and reliable assessments than absolute judgments. Recent studies with comparative evaluation support our assumption (Zheng et al. 2023;  Liusie et al. 2024). When adopting such a comparative approach for dialog evaluation, the straightforward way is to utilize multiple responses for the target context, which may not always be available. Our metric makes a detour by comparing the target response against the ones derived from other dialogue contexts.
>
> **W2**
>
> We also recognize that PairEval still requires more computational cost than the pointwise evaluation approach. However, we enabled PairEval to conduct a reliable evaluation with only a few comparisons rather than an exhaustive one. Regarding the fine-tuning cost, we kindly note that this training is only performed once before the evaluator's deployment and is not required after that. In other words, as the number of evaluated samples increases, the computational benefits from not performing exhaustive comparisons will exceed the initial fine-tuning cost.
>
> **W3**
>
> We note that human-crafted responses are required during the training process rather than at evaluation time. This benefit of PairEval eliminates the need for human-crafted data during evaluation. We will further clarify these benefits in our camera-ready version.
>
> **Q1**
>
> We believe the evaluation results on TopicalChat-USR and FED in Table 3 can address your concern, where the model trained on the DailyDailog dataset is evaluated on other datasets.
>
> **Q2**
>
> The key difference between PairEval and previous metrics lies in its comparative assessment process to elicit the final judgments. Please note the superiority of PairEval does not solely originate from adopting the large model. PairEval performs better in evaluation datasets and adversarial responses than DirectEval, which uses the same LM and different assessment processes.
>
> **Q3**
>
> The limited improvements of exhaustive comparison can be attributed to the aggregation strategy. As the number of comparisons increases, the impact of a new one on the overall result diminishes, leading to a saturated score. Our future work will devise more advanced strategies like de-emphasizing unreliable comparison outcomes.
>
> **Q4**
>
> Table 3 reports the averaged scores with different sets, while Table 6 reports the single run.

---

### Decision · Program_Chairs · 2024-07-10

**Decision:**

Accept

**Comment:**

This paper proposes a new metric for evaluating open-domain dialog quality via paired comparison to a different dialog, showing higher correlation with human judgments of quality compared to several reference-free and reference-based metrics.

The idea of an in-between evaluation metric that uses some baseline, but not from the exact conversation context, is valuable. Additionally, the reviewers found the experiments to be comprehensive. However, the main concern is that the majority of the improvement seems to come from the specific language model used rather than the pairwise design choices, which is evident in comparing directEval and PairedEval. Since the core contribution of the paper is "a new comparative evaluation paradigm," as the authors put it, this limits the applicability of this work. While results in general setups are quite mixed when comparing directEval and PairedEval, PairedEval seems to perform slightly better in an adversarial setup.